# Ultra-Low Precision 4-bit Training of Deep Neural Networks

**Xiao Sun**  **Naigang Wang**  **Chia-yu Chen**  **Jia-min Ni**

**Ankur Agrawal**  **Xiaodong Cui**  **Swagath Venkataramani**

**Kaoutar El Maghraoui**  **Vijayalakshmi Srinivasan**

**Kailash Gopalakrishnan**
IBM T. J. Watson Research Center, Yorktown Heights, NY 10598, USA
{xsun, nwang ,cchen, jiamin.ni1, ankuragr, cuix,
swagath.venkataramani, kelmaghr, viji,kailash}@us.ibm.com

## Abstract

In this paper, we propose a number of novel techniques and numerical representation formats that enable, for the very first time, the precision of training systems to be aggressively scaled from 8-bits to 4-bits. To enable this advance, we explore a novel adaptive Gradient Scaling technique (GradScale) that addresses the challenges of insufficient range and resolution in quantized gradients as well as explores the impact of quantization errors observed during model training. We theoretically analyze the role of bias in gradient quantization and propose solutions that mitigate the impact of this bias on model convergence. Finally, we examine our techniques on a spectrum of deep learning models in computer vision, speech and NLP. In combination with previously proposed solutions for 4-bit quantization of weight and activation tensors, 4-bit training shows non-significant loss in accuracy across application domains while enabling significant hardware acceleration ($>7\times$ over state of the art FP16 systems).

## 1   Introduction

Over the past decade, Deep Neural Networks (DNNs) have surpassed traditional Machine Learning techniques on a wide spectrum of application domains including speech [1–3], computer vision [4–10], and language [11, 12]. As models and datasets have grown in complexity, DNN training times have increased significantly—limiting the pace of innovation in model training and deployment life cycles.

Driven by a number of key advances in low-precision arithmetic [13–16] and the quadratic dependency of throughput on precision, reduced-precision training has become the de facto technique to boost the performance and power efficiency of deep learning hardware. 16-bit floating point training formats have been integrated into multiple accelerator products [17–19] and shown to provide 4-8 times the performance of 32-bit hardware designs. Recently, new insights into low-precision accumulation errors, batchnorm statistics and novel 8-bit floating point formats have resulted in successful demonstrations of 8-bit training for a wide range of deep learning tasks—resulting in a further $>2$-$4\times$ boost [15] in training system performance over 16-bit systems.

Meanwhile, there has been tremendous progress in the use of ultra-low precision formats (2-4 bits) for inference [20–27]. 4-bit inference [21] (where weights and activations are quantized in 4-bit integer formats) has been shown to fully preserve model accuracy and provide significant acceleration in comparison to 8-bit integer systems [28] used widely for inference [17]. Unfortunately, since approximately $2/3^{rd}$ of the training time and operations are spent on the backward GEMM and update GEMM phases, acceleration of the entire training workload will necessitate (a) representing the gradients in 4-bits formats, and (b) computing the backward and update phase entirely in 4-bits representation. To the best of our knowledge, no studies so far have demonstrated deep learning model convergence using 4-bits for all tensors (weights, activations and gradients).

While 8-bit floating point formats appear to be sufficient for training [14, 19], 4-bit gradient representations appear challenging from a quantization error (rounding), precision and dynamic range perspective. This results in significant model optimization and generalization difficulties. Furthermore, tensors used in forward and backpropagation may require dramatically different numerical representations and range. Indeed, [15] demonstrated that forward floating-point tensors require a higher resolution and a lower range in comparison to backward gradients. This issue is further exacerbated if low precision (4-bit) integers are used to represent weights and activations for inference and in general for training proposals that utilize the same number formats for all tensors (including 8-bit fixed-point[16] or the 5-bit logarithmic format[29]). These results point to a direction where successful 4-bit training will likely necessitate different numerical formats for representing different tensors in the forward and backward phases of training.

In this work we demonstrate, for the very first time, an end-to-end solution that uses 4-bits for the vast majority of the computations needed during DNN training. We propose a novel 4-bit floating point format, rounding schemes, as well as new gradient scaling techniques that minimize gradient representation, precision and range challenges. These methods enable model convergence with negligible accuracy loss for a spectrum of deep learning benchmarks. In addition, instead of using the same 4-bit gradient format to represent weights and activations, we integrate state-of-the-art 4-bit inference methodologies alongside our techniques. Furthermore, we note that the design of our 4-bit training solution is guided by generations of deep learning hardware design expertise as well as compiler driven performance optimizations.

## 2    Related Work, Challenges and Key Contributions

Banner et al. [16] presented low-precision training results with 8-bit fixed-point integers for the forward phase as well as a subset of the computations in the backward phase. While fixed-point formats represent weights and activations very well, floating point formats are advantageous for gradient representation due to their large dynamic range. At the other end of the spectrum, Miyashita et. al. [29] proposed a training scheme based on pure logarithmic 5-bit representations and demonstrated good results on the AlexNet model (on CIFAR10). Larger datasets and models using pure logarithmic representation have not shown good results and are likely limited by the higher representation needed for weights and activations [15]. [14, 15] present a number of key ideas to enable 8-bit floating point (FP8) training including (a) Novel Hybrid FP8 formats to represent weights, activations and gradients (b) Chunk-based hierarchical accumulations to minimize low-precision accumulation errors (c) Selective precision rules (for first, last and depthwise convolutional layers) and (d) Automatic Loss Scaling Approaches (APEX) [19, 30]. These techniques have allowed FP8 trained DNN models to achieve <1% accuracy degradation (vs. FP32 baselines) across datasets.

While the best training results are achieved using FP8, a number of new ideas have enabled inference precision to be scaled aggressively to <4-bit integer representations using quantization-aware training techniques [20–22, 25]. Choi et. al. [21] introduced PACT (Parameterized Clipping Activation) to automatically learn the clipping range for activations during training. They also proposed a statistics aware weight binning (SAWB) technique to quantize weights based on the L1 and L2 norms [22] and full precision short cuts (FPSC) in residual networks to achieve 2-bit inference with just 2-3% degradation on large ImageNet based models. Recent work [20–27] has improved on these techniques by using a whole host of new ideas including trainable weight scaling factors. While 2-bit quantized models still suffer >2% accuracy, 4-bit networks seem to converge well without any major loss in model fidelity. In this work, we build on the PACT, SAWB and FPSC techniques to quantize weights and activations in 4-bit integers (WA-INT4) and combine them with new gradient formats and quantization methods. 4-bit training where gradients, weights and activations are all represented using

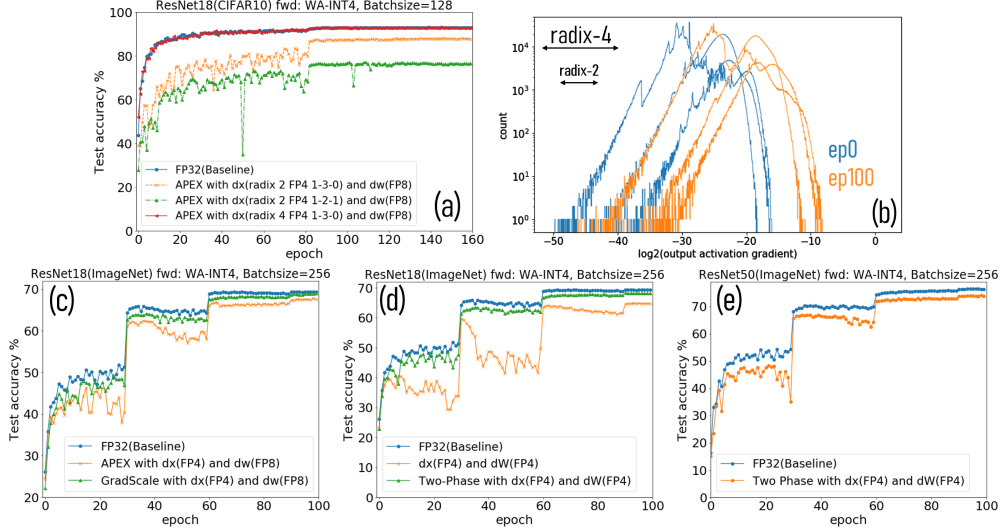

Figure 1: 4-bit training challenges : (a) ResNet18 (CIFAR10) model convergence with FP32 (baseline) and with gradients in radix-2 (1-3-0), radix-2 (1-2-1) and radix-4 (1-3-0) floating point formats and APEX loss scaling (while using INT4 for weights and activations, WA-INT4). Radix-4 formats are critical to maintain dynamic range. (b) Gradient distribution of specific layers in ResNet18 (ImageNet) at epoch 0 and epoch100. Different layers show widely different ranges of gradients across epochs. (c) ResNet18 (ImageNet) model accuracy with partial backpropagation in FP4 (and the rest in FP8) while using loss-scaling(APEX). APEX causing degradation from the baseline due to insufficient range vs. our proposed technique (GradScale) which shows excellent convergence. (d) ResNet18 (ImageNet) test accuracy for full FP4 backward (dx and dW) with traditional and the proposed two-phase FP4 rounding to minimize bias and internal covariate shifts. (e) ResNet50 (ImageNet) convergence challenges with 2-phase rounding attributed to the FP4 gradient variance reduction when passing through the Conv 1x1 layers.

4-bits has a number of key challenges. Using ResNet18 (CIFAR10), we explore the combination of using INT4 for weight and activation tensors along with various FP4 formats for gradients as shown in Fig. 1(a). For the forward path, we use PACT and SAWB techniques for INT4 activation and weight quantizations respectively. Using the same loss scaling technique, APEX [30], we observe that the radix-2 FP4 formats ([sign,exponent,mantissa] = [1,2,1] and [1,3,0]) suffer from 10-30% accuracy loss when being applied on the backward GEMM function($dx$) (the update GEMM function ($dW$) remains in FP8). The primary reason for this lack of convergence is the limited dynamic range ($= 2^3$–$2^6$) when using the radix-2 formats for gradients. In contrast, the radix-4 (1,3,0) format extends the dynamic range from $2^6$ to $2^{12}$ and facilitates excellent convergence—underlying the importance of the radix-4 FP4 format for 4-bit training. Going forward, in the remaining sections, the FP4 format (unless explicitly indicated) will be assumed to be radix-4 and will be described in detail in Section 3.1.

While ResNet18 (CIFAR10) shows good convergence with FP4, the dynamic range of gradients is significantly higher in models using the ImageNet dataset (as shown in Fig. 1(b)). We note that different layers exhibit widely different gradient ranges and therefore a central loss/gradient scaling technique (like APEX[30]) will not work well for FP4 gradient representation across all layers. This is further validated in Fig. 1(c), where we observe significant degradation (> 1.5%) for ResNet18 (ImageNet) while using APEX with FP4. This effect is noticeable even in cases when just **dx** is computed using FP4 while **dW** remains in FP8. To overcome this accuracy loss, we propose GradScale, a new per-layer gradient scaling technique (discussed in Section 3.2) that improves accuracy to within ~0.5% of the FP32 result.

Furthermore, when **both** backward and update GEMM computations of ResNet18 (ImageNet) are quantized using FP4, we observe significant loss in model fidelity (>4.5%) as noted in Fig. 1(d). In Section 4.1, we analyze the root causes of this loss in accuracy and attribute it to the bias introduced during FP4 rounding which in turn causes large internal covariate shifts. Hardware friendly solutions to this rounding problem (referred to as Two-phase rounding) are proposed in Section 3.3 and enable full 4-bit trained models to recover to within ~1% of the FP32 baseline.

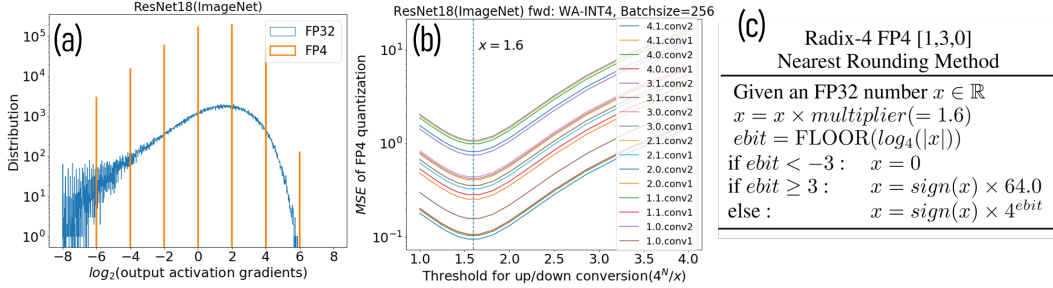

Figure 2: (a) Gradients of the last convolution layer of ResNet18 (ImageNet) overlaid with the radix-4 FP4 format—illustrating that radix-4 (unlike radix-2) has sufficient range to represent most gradients. (b)Mean Squared Error (MSE) of FP4 tensors vs. thresholds (for quantization). As expected, the midpoint of the $[4^{n-1}, 4^n] = 4^n/1.6$ minimizes MSE. (c)the FP4 nearest rounding procedure

Finally, in DNN models with higher complexity including ResNet50 (Fig. 1(e)) on ImageNet (vs. smaller models like ResNet18), we observe much higher accuracy degradation (2.5%) in spite of exploiting all of these techniques. This is strongly correlated to the diminishing gradient variances due to the aggressive 4-bit quantization. Using ablation studies, we propose solutions to mitigate this impact by using 8-bit gradients on the critical layers for accuracy while retaining the majority of the benefits of 4-bit computations.

To summarize, this work enables **end-to-end 4-bit DNN training** and our key contributions include:

1. A new **radix-4 FP4** format that enables representation of gradients with a wide dynamic range.
2. A new per-layer trainable gradient scaling technique, **GradScale**, that aligns gradients to the FP4 range—essentially maximizing the use of the range for each layer and circumventing the challenges of using a global loss scaling factor (APEX) across all layers.
3. A **Two-Phase** quantization technique to minimize the quantization error of FP4 gradients for both the mean squared and expected errors for multiple uses of the same gradient—achieving ~1% accuracy degradation on CNN models on the CIFAR10 and ImageNet dataset.
4. A deeper understanding of the impact of **quantization bias** on training and its interplay with batch normalization. We show evidence that aggressive quantization causes internal covariate shifts that result in a generalization problem.
5. A **hybrid** approach exploiting FP4 precision for a majority of the computations and FP8 for a minority of the computations in the bottleneck—such as the Conv1x1 layers in ResNets—to mitigate the latter's impact on gradient variances, resulting in further improvement of model accuracy for deep models including ResNet50.
6. Experiments combining gradients represented using FP4 with state-of-the-art INT4 techniques for weights and activations demonstrating high accuracy across a suite of models and datasets.

## 3   Key Technical Contributions

### 3.1   Radix-4 FP4 Format

The radix-4 FP4 format with [sign,exponent,mantissa] = [1,3,0] is essentially a logarithmic format $(4^n)$ that spans a range of $\pm 4^3 (= \pm 2^6)$ and can represent (scaled) gradient values as small as $4^{-3}(= 2^{-6})$ (Fig. 2(a)). Given the limited number of bits, we follow floating point arithmetic guidelines to preserve the all-zero binade to represent $\pm 0$ but skip NaN and infinity representations. We use nearest rounding schemes and choose the mid-point $((4^n + 4^{n-1})/2 = 4^n/1.6)$ between two neighbouring exponent levels $([4^{n-1}, 4^n])$ as the rounding threshold, as shown in Eq.1. This rounding threshold helps minimize the quantization mean square error (MSE) for each gradient tensor. As shown in Fig. 2(b) for each layer of ResNet18 (ImageNet), the smallest MSE is obtained when a threshold of $4^n/1.6$ is chosen for final FP4 gradient quantization.

$$round(x) = \begin{cases} 4^{n-1} & x \leq 4^n/1.6 \\ 4^n & x > 4^n/1.6 \end{cases} \qquad (1)$$

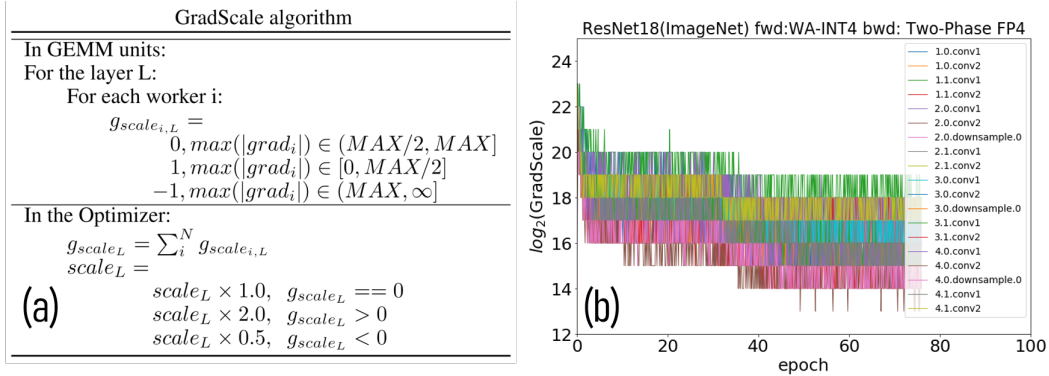

Figure 3: GradScale—a per-layer trainable scaling factor: (a) GradScale Update Algorithm (b) Scaling factor for FP4 gradients for each layer of ResNet18(ImageNet). As can be seen, different layers exhibit widely different ranges—so a layer-wise factor is critical to prevent overflow and underflows.

The conversion from higher precision formats to FP4 can be very efficiently implemented (both in software and hardware) as shown in Fig. 2(c). To summarize, higher precision floating point numbers within the FP4 dynamic range are multiplied by 1.6 (to enable nearest rounding) and the exponent bits are then directly chosen via truncation for FP4 representations.

## 3.2   GradScale: Trainable Layer-wise Gradient Scaling

For low-precision (FP16 and FP8) training, loss scaling approaches are widely used to scale up the loss before backpropagation in order to align the maximum value of the gradients to the maximum representable value in the used precision format [19, 30]. However, since the scaling factor is directly applied to the loss, the gradients in all the layers of the deep network share the same scale. As discussed in Section 2 (Fig. 1(b) and 1(c)), this causes a significant range problem for FP4 representations due to the diversity in the gradient values across layers. To extend loss scaling to each individual layer in an automatic and dynamic manner, we propose a new technique called GradScale (Fig. 3 (a)) that scales up the output gradients of each layer before Backward and Update GEMMs for input gradient and weight gradient computations respectively.

To implement GradScale with minimal changes to the original DNN graph, we treat per-layer gradient scaling factors as trainable parameters. We then update these parameters based on a binary "gradient" dependent on whether the maximum gradient in each layer falls within $[2^5, 2^6]$—i.e. part of the highest bin in the FP4 format. Each learner (in a multi-GPU training system) reports "overflow" if the maximum gradient is greater than $2^6$ after scaling, and "underflow" if smaller than $2^5$ per layer. Based on the reports from each learner, the optimizer will either preserve, double or halve the scaling factor for each layer, as shown by a weight-update-like process described by the algorithm in Fig. 3(a). It should be noted that for non i.i.d. data, each worker could in principle have its own per-layer scaling factor—but for i.i.d data we see no difference and thus simply add the trainable parameters across learners to avoid revising any communication protocols. A consequence of per-layer scaling is that the output of backward GEMMs for each layer needs to be scaled down before being sent to the optimizer and previous layers in a DNN—which is more difficult to manage than loss-scaling in APEX. Details of how the scaling factors are applied to the 3 GEMMs (and how it could be optimized in hardware / software) are discussed in Appendix-A. As shown in Fig 3(b), the optimized scaling factor for different layers of ResNet18 (ImageNet) when using FP4 exhibits as much as $2^{5-6}$ difference, consistent with its FP32 counterpart in Fig. 1(b). Furthermore, we clearly see that, using trainable parameters, GradScale is able to capture the gradients of all the layers effectively enabling FP4 quantization—unlike APEX and other global loss scaling techniques. Finally, as shown in Fig.1(c), the accuracy for ResNet18 (ImageNet) improves dramatically (> 1% vs. APEX) when using GradScale and is within 0.5% of the FP32 baseline.

### 3.3 FP4: Two-Phase Rounding (TPR)

As explained in Fig. 1(d), when FP4 is used in both the backward and update GEMMs, we observe significant accuracy degradation (> 4.5%). To improve this accuracy loss, we propose a novel Two-phase Rounding (TPR) technique by defining two sets of radix-4 FP4 formats, the original FP4-even format that has $2^{even}$ values at each level and a new FP4-odd format that has $2^{odd}$ values at each level, as shown in Fig. 4 (a). Given a gradient, we adopt the appropriate mid-points from Fig. 2 (c) and quantize it to both the even and the odd phases—resulting in an expected quantization error that will be mitigated due to cancellation as shown in Fig. 4 (a). The reason for considering two quantizations for the same gradient is based on the fact that each output activation gradient $dL/dy$ is used twice during back propagation: one for calculating the input activation gradient $dL/dx$ ($dL/dx = dL/dy.W^T$) and the other for calculating the weight activation gradient ($dL/dW$ in $dL/dW = x.dL/dy$). This enables us to "look twice" at the same gradient with two sets of filters (quantizers). It should be noted that in our experiments, we adopt the even phase for $dL/dx$ and the odd phase for $dL/dW$. In other words, having two different quantizations of $dL/dy$ allows us to retain more information on the gradients (even though they're used in different computations). Furthermore, our experiments show that the reverse case using the FP4-even phase for $dL/dW$ and the FP4-odd phase for $dL/dx$ results in similar convergence results on ResNet18 (ImageNet). The mean of the TPR FP4 quantized numbers approaches the quantized value obtained under the radix-2 format—which has twice the representation power over the radix-4 format. As shown in Fig. 1(d), the TPR technique significantly improves the test accuracy (by >3.5%) in comparison to the basic FP4 rounding scheme and recovers the final model accuracy to within ~1% for end-to-end 4-bit training of ResNet18. A summary of the forward, backward and update GEMM computations used in end-to-end 4-bit training is shown in Fig.4 (b). In the forward path, INT4 based multiply-accumulate (MAC) engines are used while in the backward phase INT4×FP4 computations need to be supported. From a hardware perspective, these computations are significantly more efficient than traditional FP16 or FP8 designs. Leveraging insights derived from multiple generations of low-precision floating point unit (FPU) designs, we've estimated that MAC engines with INT4×FP4 capabilities are >7× more area and power efficient in comparison to FP16 MAC engines. A detailed discussion of these capabilities can be found in Section 6.

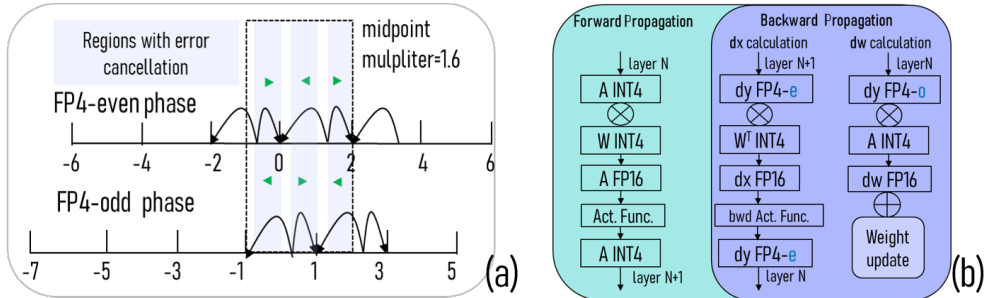

Figure 4: (a) Two Phase Rounding demonstrating cancellation of FP4 errors. The x-axis are values in log2 base, the green arrows indicate the direction of rounding (i.e. up or down) for the numbers in the colored region on the axis. When the green arrow directions are opposite between the FP4-even and the FP4-odd phases, the quantization errors in the two phases cancel each other. In other areas, each phase has a 50% chance of yielding a smaller quantization error than the other. (b)The summary of precisions used in our final 4-bit training setup using INT4 for weights and activations and FP4 for gradients(-**e**ven or -**o**dd). A Mixed Precision INT4-FP4 Multiple Accumulate (MAC) Engine is needed for backward and update GEMM computations.

## 4 Discussion of Key Results

### 4.1 Gradient Quantization Induced Internal Covariate Shift

In order to understand some of the key results and the benefits of TPR, we explore the impact of FP4 quantization errors ($q_{error}$) on the expectation value of the weight gradients $dL/dw$:

$$E[dL/dw] = E[x \cdot (dL/dy + q_{error})] = E[x \cdot dL/dy] + E[x \cdot q_{error}] \qquad (2)$$

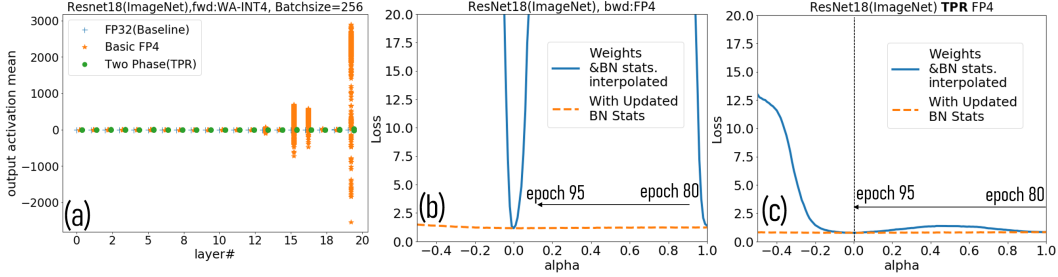

Figure 5: (a) The mean $\mu_B$ for each batchnorm layer following a convolution layer at epoch 95 for FP32, basic FP4, and TPR FP4 for ResNet18 (ImageNet). Loss landscape with interpolated parameters and $\mu_B$ & $\sigma_B$ between the model in epochs 80 and 95 for (b) 1-phase FP4 and (c) TPR FP4 quantization. Also shown in (b) is the landscape with $\mu_B$ & $\sigma_B$ updated in the interpolated space for the basic FP4 quantization scheme.

Since input activations ($x$) are always positive after ReLU and E[$q_{error}$] tends to be non-zero for typical gradient distributions (please refer to Appendix-B), the 2nd term in Eqn.2 will introduce a non-zero bias. This gradient bias will skew weight tensors during training which in turn may cause a large shift in output activations, i.e. an internal covariate shift (ICS) that is induced by gradient quantization, as shown in Fig. 5(a). Interestingly, even with a large ICS, the training process can still converge due to batchnorm layers. Santurkar et. al. [31] demonstrated that instead of suppressing ICS, batchnorm layers adjust mean and variance to compensate its effects. However, relying on batchnorm corrections is challenging due to the strong dependence of loss/accuracy on the batch's mean $\mu_B$ and standard deviation $\sigma_B$. We note that normalized activation, as it passes through the batchnorm layers, can be represented as $\hat{x} = \frac{x-(\mu_B+\Delta\mu_B)}{\sigma_B+\Delta\sigma_B}$, where $\Delta\mu_B$ is the mismatch among samples and is usually a tiny fraction of $\mu_B$. The evaluation on testing data will fail if $|\Delta\mu_B| >> |x-\mu_B|$. This is clearly visible in the loss landscape along the direction of optimization during training as shown in Fig. 5(b)—where we plot the landscape [32, 33] on the interpolated 1D space of all parameters (including $\mu_B$ & $\sigma_B$) in epochs 80 and 95. Sharp minima in the losses are observed around the optimization points—which can be recovered into a flatter landscape (dashed curve) if $\mu_B$ & $\sigma_B$ are updated at each point of the interpolated space. This confirms our theory that batchnorm layers correct gradient-quantization induced ICS causing losses that are highly dependent on $\mu_B$ and $\sigma_B$ in those layers. Since $\mu_B$ and $\sigma_B$ in the testing data may slightly mismatch their training values, this results in poor generalization. Two-phase rounding schemes can mitigate this quantization error and greatly suppress the ICS as shown in Fig. 5(a). Using TPR, the local minima in epochs 80 and 95 in Fig. 5(c) also become much flatter.

## 4.2 Ablation Studies on Restoring Gradient Variances

Ablation studies, using the variance of the input gradients Var($dL/dx$) as a metric, were used to analyze the impact of FP4 quantization on various layers of a DNN. We choose Var($dL/dx$) because of the recognized importance of this variance in DNN optimization and generalization [34–37]. In Fig. 6(a), for ResNet50 (ImageNet), we observe that the reduction of Var($dL/dx$) from the baseline becomes larger as the TPR FP4 gradients backpropagate through the network, indicating diminishing gradient variances with layer depth due to 4-bit training. Interestingly, the bottom 1x1 convolutional layers in each residual block appear to have the highest impact on Var($dL/dx$) and when both 1x1 layers are skipped, the variance is mostly restored. These results provide us clear guidelines on how to apply FP8 selectively to specific layers in 4-bit training systems as shown in Fig. 6(b). As an example, when bottom 1x1 layers (or both 1x1 layers) in each residual block are set to FP8 (while using TPR FP4 everywhere else), accuracy improves by 1.0% (1.5%)—while still providing 5.2× (4.0×) training acceleration over FP16 systems (Fig. 6(c)).

## 5 Experimental

To demonstrate the robustness of the proposed 4-bit training scheme, we examined the impact of using INT4 weights and activations and FP4 gradients on a spectrum of computer vision models on the CIFAR10 [38] and ImageNet [39] datasets, as summarized in Tables 1 and 2 respectively. These

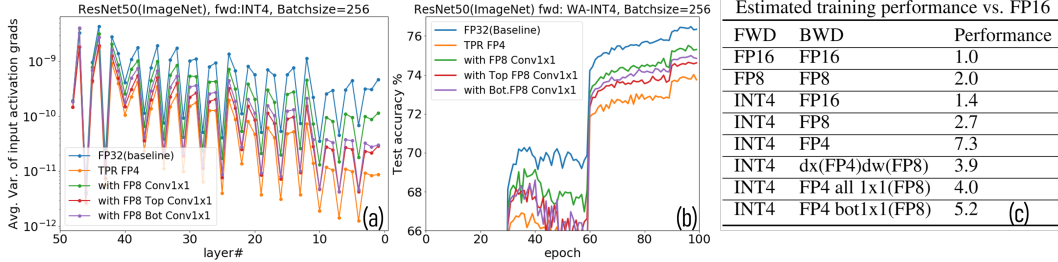

Figure 6: Ablation Insights: (a) Averaged variances of the input activation gradients for each layer of ResNet50(ImageNet) vs. layer precision. (b) ResNet50 (ImageNet) convergence using these insights. (c) Performance estimations of various INT4/FP4/FP8/FP16 training solutions (details in Appendix-C).

emulation results were performed using a custom-modified PyTorch framework that implemented all of the precisions and schemes discussed in the paper (details in the Appendix-A). For all of these models, we used default network architectures, pre-processing techniques, hyper-parameters and optimizers with 4-bit training. For CIFAR10, using TPR FP4 alone, all models without 1x1 convolutional layers achieve < ~0.5% accuracy degradation (vs. FP32 baselines) but deeper models including MobilenetV2, ResNet50 and ResNet101 [9] showed slightly > 1% accuracy losses. Utilizing the gradient variance insights of Section 4.2, ResNets are able to achieve accuracies within ~0.5% when the Conv 1x1 layers are computed in FP8. MobileNetV2 is more difficult to train in 4-bit due to lesser redundancy in its parameters[15]. Additional venues for optimization and improved accuracy are also shown in Tables 1 and 2 and include selectively applying FP4 to the backward GEMM while utilizing FP8 gradients for the update GEMM. With this FP4/FP8 hybrid backward technique, MobileNetV2 can achieve 0.6% accuracy loss while still retaining 3.9× hardware performance enhancement in GEMM (Fig.6c). It should be noted that we follow the insights in [15] to use FP16 for depthwise layers, which consume <3% of total ops and are difficult to accelerate due to limited data reuse.

On ImageNet, the full TPR FP4 technique achieved ~1% accuracy loss for AlexNet [5] and ResNet18. For larger models on ImageNet such as ResNet50, the TPR FP4 scheme shows 2.5% degradation—which can be recovered to <1% (or ~1.5%) by applying FP8 on all (or bottom) 1x1 convolutional layers respectively. These results are consistent with what we've observed on CIFAR10. On ImageNet, we also observe in ResNet50 and MobileNetV2 that INT4 in the forward path by itself causes 0.7% and 2.0% loss in accuracy respectively—which we expect can be significantly improved using recently published techniques [20, 23].

In addition, looking beyond CNNs, we've applied TPR FP4 techniques to Speech and NLP models as shown in Table 3. In NLP, we've achieved excellent results on the PTB dataset with negligible loss in perplexity. For larger networks, we achieved only ~2% loss in BLEU score for the Transformer base model on the WMT En-De task. In speech, TPR FP4 shows ~1.4% degradation in Word Error Rate (WER) when training a 4-layer Bidirectional LSTM model on the SWB300 dataset [40], and < 0.5% WER loss when we use FP8 for dw (update GEMM) computations. Details on all of these models and final convergence curves can be found in Appendix-D. These results mark the first experimental demonstrations of 4-bit training—showing robustness across a spectrum of models and datasets with limited loss in accuracy and significant gains in performance (Fig. 6(c)).

# 6 Hardware Design Cost and Overhead

The evaluation of hardware cost for 4-bit training is based on insights derived from multiple previous AI Hardware and Low-precision FPU designs [41, 42]. In our estimates, a MAC unit that performs 4-way INT4×FP4 inner products to support 4-bit backpropagation consumes 55% of the area of the FP16 FPU while providing 4× throughput, yielding a total compute density improvement of 7.3×. Compared to FP16 FPUs, the 4-bit unit has simpler shift-based multipliers thanks to the power-of-2 FP4 numbers. It also benefits from the absence of addend aligners, narrower adders, and a simpler normalizer.

The implementation of TPR has negligible hardware design cost. The conversion of gradients to FP4 formats in the even and odd phases can be implemented using the same conversion unit. In addition, dedicated MAC units are not needed to handle GEMM with FP4 operands in the additional odd phase. FP4 numbers in the odd phase represent values simply $0.5\times$ of those in the even phase. Thus, the odd-phase final MAC result is also $0.5\times$ and can therefore be efficiently computed by subtracting 1 from the exponent of the final FP16 result using the same MAC engine.

The overhead needed to enable 4-bit training is also small and is estimated to be <5% of the total GEMM ops. In back-propagation, specifically, we expect to see ~10 FP16 FLOPs/gradient for PACT backward, inter-radix conversion, TPR, and Layer-wise scaling overheads. These overheads are much smaller than $\mathcal{O}(k_i \times k_j \times channel)$/gradient in convolution GEMMs (e.g. In ResNet50, the effective GEMM FLOPs is 642 for each gradient). With the majority of FLOPs spent on GEMM computations, 4-bit training is expected to retain significant advantage over 8-bit training due to throughput, power and area gains in going from 8-bit to 4-bit GEMM.

Table 1: CIFAR10 test accuracies using 4-bit training

| FWD | FP32 | INT4 | INT4 | INT4 | INT4 | INT4 |
|---|---|---|---|---|---|---|
| BWD | FP32 | FP32 | TPR FP4 | dx(FP4) dW(FP8) | TPR FP4 bot 1x1(FP8) | TPR FP4 1x1(FP8) |
| VGG16 | 92.07 | 91.81 | **91.47** | 91.81 | - | - |
| GoogLeNet | 94.68 | 94.50 | **94.17** | 94.12 | - | - |
| ResNet18 | 93.01 | 92.97 | **92.74** | 92.76 | - | - |
| ResNet50 | 94.70 | 94.34 | 93.42 | 93.77 | 94.03 | **94.36** |
| ResNet101 | 95.04 | 94.76 | 93.79 | 94.34 | 94.19 | **94.43** |
| DenseNet121 | 95.06 | 95.08 | **94.71** | 94.82 | - | 94.91 |
| MobileNetV2 | 93.00 | 92.97 | 91.80 | **92.39** | - | - |

Table 2: ImageNet test accuracies using 4-bit training

| FWD | FP32 | INT4 | INT4 | INT4 | INT4 | INT4 |
|---|---|---|---|---|---|---|
| BWD | FP32 | FP32 | TPR FP4 | dx(FP4) dW(FP8) | TPR FP4 bot1x1(FP8) | TPR FP4 1x1(FP8) |
| Alexnet | 57.56 | 57.51 | **56.38** | 57.11 | - | - |
| ResNet18 | 69.40 | 69.43 | **68.27** | 68.99 | - | - |
| ResNet50 | 76.48 | 75.76 | 74.01 | 74.92 | 74.99 | **75.51** |
| MobileNetV2 | 71.85 | 69.77 | 68.85 | 69.65 | - | - |

Table 3: NLP&speech accuracies w/4-bit training

| FWD | FP32 | INT4 | INT4 |
|---|---|---|---|
| BWD | FP32 | TPR FP4 | dx(FP4) dW(FP8) |
| 2-layer-LSTM (PTB)Test ppl. | 83.3 | **85.3** | 83.8 |
| 4-layer-biLSTM (SWB300)WER | 9.9 | 11.3 | **10.4** |
| Transformer-base (WMT En-De)BLEU | 27.5 | 25.4 | 25.9 |

# 7   Conclusion

For the very first time, we've demonstrated successful 4-bit training on a variety of deep learning benchmarks with limited accuracy losses. To accomplish these results, we use quantization-aware training techniques for INT4 weights and activations along side the newly proposed FP4 numerical format for gradients. To enable end-to-end 4-bit training, the radix-4 FP4 formats are combined with new Two-Phase rounding (TPR) and Gradient Scaling (GradScale) techniques that allows us to maximize the range and representation needed for gradients. Finally, we analyze the key factors that impact accuracy in 4-bit training systems including Internal Covariate Shifts and Diminishing Gradient Variance and propose solutions that improve model fidelity while retaining most of the benefits of 4-bit training. In addition, we expect training performance incorporating FP4 to be 4-7× faster in comparison to current FP16 designs. Future work directions include further improving accuracies for both 4-bit forward and backward propagations in large and compact models, such as Transformer and MobileNets, and broadening the scope of DNN models and datasets to cover additional domains including object detection.

## Acknowledgments

This work is fully funded by IBM Research. The authors would like to thank Anthony Giordano, I-Hsin Chung, Ming-Hung Chen and James Norris for their helpful support on computing infrastructure, and Jungwook Choi, Sunil Shukla, Leland Chang, Arvind Kumar, Yulong Li, Jie Yang, Jinwook Oh, Mingu Kang, Marcel Schaal, Mauricio Serrano, Wei Wang and other current/previous members in the IBM AI accelerator team for the chip platform targeted in this work. The authors would also like to thank Jeffrey Burns, Mukesh Khare and Tze-chiang (T.C.) Chen for executive support & leadership, and IBM AI Hardware Center for computing resources that have significantly accelerated the learning. This work is realized by generous collaborations across IBM Research.

## Broader Impact

Dedicated hardware accelerators for DNN training, including GPUs and TPUs, have powered machine learning research and model exploration over the past decade. These devices have enabled training on very large models and complex datasets (necessitating 10 - 100's of ExaOps during the training process). Reduced precision innovations (16-bits) have recently improved the capability of these accelerators by 4-8$\times$ and have dramatically improved the pace of model innovation and build. The 4-bit training results, presented in this work, aim to push this front aggressively and can power faster and cheaper training systems for a wide spectrum of deep learning models and domains. To summarize, we believe that 4-bit training solutions can accelerate ML research ubiquitously and provide huge cost and energy savings for corporations and research institutes—in addition to helping reduce the carbon / climate impact of AI training. By improving the power efficiency by $4 - 7\times$ in comparison to current FP16 designs (and $> 20\times$ vs. default FP32 designs), the carbon footprint for training large DNN models can be significantly reduced [43].

The reduction in computational energy and memory footprint needed by 4-bit training systems could also enable training to be carried out on edge devices (e.g. mobile platforms). This, in turn, could alleviate security and privacy concerns of sending data back to the Cloud for aggregated AI model build.

We would also like to point out that, although we show promising results and limited accuracy loss in comparison to FP32 training, 4-bit training solutions could still be subject to training instabilities. This may necessitate a careful examination of these training techniques over a wider range of models and perfected alongside the development of ML model research. The risk of using 4-bit training for DNN model build in real applications is higher than 8-bit or 16-bit approaches and thus requires task-specific robustness studies.

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
