[Supplementary Material]

# Appendix A    Pytorch Emulation Environment

To accurately emulate the data precision (weights, activations and their gradients) and the partial-sum precisions in the accumulator of hardware, we revise the Pytorch[1] GEMM GPU Kernels to quantize the data and accumulation at a single FLOP level. A typical graph with the original layers, forward/backward quantization layers, and GradScale layers is shown in Fig.s-1, which resembles the hardware implementation in Fig.s-2.

**Wrapped Convolution layer for 4-bit training Emulation**

Fig.s- 1.  the flow chart of quantization and GradScale emulated in Pytorch framework, which resembles the hardware implementation in Fig.s-2, in which $S_{FP}$, $S_x$, and $S_W$ are scaling factors for gradients, activations and weights, and Q () is a quantization function.

Fig.s- 2.  a designed hardware implementation of 4-bit training, in which, $x$, $y$, and $W$ are input activations, output activatoins and weights, $dx$, $dy$, and $dW$ are their gradients, $S_{FP}$, $S_x$, and $S_W$ are scaling factors for gradients, activations and weights, and Q () is a quantization function.

We apply this quantization wrapper on each layer of the network except the first and the last layer. Following the conventions of DNN quantization[2–7], we keep the first Conv layer and the last fully-connected (FC) layer in a higher FP16 (1-6-9) precision to maintain the input and output fidelity—which constitutes a small proportion ($< 3\%$) of the total computation complexity. To enhance the flow-back of gradients for ResNets, we also adopt the full precision shortcut (FPSC) approach [8] by using FP16 (1-6-9) for the Conv 1x1 layers on the shortcut path, which amounts to $< 1\%$ of the total computation.

## A.1    Forward

To emulate 4-bit INT4 and FP4 convolution, we quantize the activations and weights to INT4 and recover it to FP32 format for GPU computation, which we refer to as fakeI4 ($fI4$).

$$\mathbf{x}_{fI4} = \mathbf{x}_{I4}.S_x \qquad (1)$$
$$\mathbf{W}_{fI4} = \mathbf{W}_{I4}.S_W \qquad (2)$$

Following [7, 8]'s promising inference results, we adopted PACT (Parameterized Clipping Activation) to quantize the activations, and SAWB (statistics aware weight binning) to quanitze the weights separately. Both PACT and SAWB use standard uniform (evenly spaced) quantization scheme with nearest-rounding, where SAWB is symmetrical around zero and PACT can be symmetrical or positive depending on whether the activations pass non-linear functions like ReLU before the GEMMs. The scaling factors, $S_x$ and $S_w$, define the largest quantization levels. For PACT, $S_x$ is a learnable parameter optimized during training, whereas $S_w$ is obtained from the first and second momentum of weight tensors factored by coefficients extracted from six standard distributions. More details on PACT and SAWB can be found in [7, 8].

## A.2  Backward Quantization

Following the backpropagation direction, the output gradient tensor $\mathbf{dL/dy}$ (referred to as $\mathbf{dy}$ herein) is firstly scaled by the GradScale unit (the scaling-up layer in Fig.s-1) then quantized to FP4 format. In the Pytorch emulation, it will be stored in FP32 format but with FP4 precision, following the rounding method in Fig.2 (c) in the paper. For the two phase rounding in Section 3.3, the scaled $\mathbf{dy}$ will be rounded with FP4_even and FP4_odd phases at the backward GEMM and update GEMM, respectively.

$$\mathbf{dy}_{FP4} = Q_{FP4}(\mathbf{dy}.S_{FP}) \tag{3}$$

## A.3  Backward GEMM

During backward GEMM, the scaled-then-quantized output gradient $\mathbf{dy}_{FP4}$ is convoluted with the transposed weight in fake-INT4 format in our quantized Pytorch kernel with the partial-sum and output in FP16 (1-6-9) precision:

$$\mathbf{dx}_{FP16} = \mathbf{dy}_{FP4}\mathbf{W}_{fI4}^T \tag{4}$$

It should be noted that the input gradient $\mathbf{dx}_{FP16}$ needs to be scaled back by $1/S_{FP}$ before leaving the current Conv layer. In our software emulation, this is automatically done by the backward pass of the scaling-down layer in Fig.s-1 as part of the auto-grad process of Pytorch.

## A.4  Update GEMM

Unlike backward GEMM, the output of update GEMM will exit the backpropagation and enter the optimizer to update weights, therefore it will not pass the scaling-down layer to be scaled-back by $S_{FP}$. To avoid manually scaling back the weight gradients $\mathbf{dW}_{FP16}$, we instead scale down the activations $\mathbf{x}_{fI4}$ in the forward pass of the scaling down layer in Fig.s-1. As such, the $1/S_{FP}$ in the activation will automatically scale back the weight gradient when it is calculated, as shown by the Eqn. below. Please note that this does not change the emulated results because the INT4 quantization is "agnostic" to scaling. In the hardware, it is equivalent to dividing the INT4 scaling factor $S_x$ by $S_{FP}$ (Fig.s-2). It will not change the forward output either as when the activations pass the scaling-up layer in Fig.s-1, their scaling factor will be recovered to the original $S_x$ by being multiplied with $S_{FP}$.

$$\mathbf{dW}_{FP16} = \frac{\mathbf{x}_{fI4}\mathbf{dy}_{FP4}}{S_{FP}} \tag{5}$$

## A.5  Demonstration of a 4-bit Conv

Using a toy example of Conv 3x3, we demonstrate that the weights and activations are quantized to INT4 precisions in Fig.s-3. We also show that, compatible with Pytorch's auto-grad mechanisms, we can control the precision of all the gradients in Fig.s-4.

## A.6  Training models in 4-bit

The design in Fig.s-1 enables a fast transformation of FP32 DNN models for 4-bit training. We apply the scaling-down, INT4 Quant, FP4 Quant, and scaling-up layers as a wrapper around the original GPU Conv layers (or FC layers). As an example, Fig.s-5 shows the quantized model file with its layers assigned to different forward and backward precisions for training ResNet50 (ImageNet) partially in 4-bit.

**Define FP4 and FP32 Conv 3x3**

**FWD: INT4 W and A**
**BWD:**
**backward GEMM: FP4-even**
**update GEMM: FP4-odd**
**(output prec set to low for demo purpose)**

```
conv4 = Conv4bit(in_channels=1, out_channels=1, kernel_size=3, bias = None,
        #FWD: W4A4, and PACT init alpha
                num_bits_weight = 4,
                num_bits_feature = 4,
                act_clip_init_val=64.0 ,
        #BWD: bwd GEMM: FP4 even, update GEM: FP4 odd
                g_data = [23, 8, 0, 3, 1, 3],
                g_fma =  [23, 8, 9, 6, 9, 6]).cuda()
# FP32
conv32= nn.Conv2d(1,1,(3,3),bias=False).cuda()
# set initial weights equal for FP32 and FP4 layers
conv32.weight = Parameter(FP32_weight)
conv4.weight = Parameter(FP4_weight)
```

**Print out input and weight of FP32 and FP4 layers**
**Note that fake-INT4 numbers have limited levels**

```
FP32_out = conv32(FP32_in)
print('FP32_input \n {}'.format(FP32_in))
print('FP32_weight \n {}'.format(FP32_weight))
print('-'*100)
# print out weight and input inside the forward function
FP4_out = conv4(FP4_in)
```

```
FP32_input
 tensor([[[[ 2.9157,  1.3996, 15.5272, 26.9969,  4.1042],
           [14.3333,  2.1545,  4.1251,  1.2565, 15.3056],
           [ 2.2931,  1.4201,  1.1589,  3.4858,  2.6755],
           [ 8.8990,  4.0600,  4.6695,  5.2786,  3.0775],
           [ 4.2508,  3.4396,  7.9922,  1.0452,  2.1524]]]],
       device='cuda:0', requires_grad=True)
FP32_weight
 tensor([[[[ 0.5756,  0.0220, 38.8300],
           [ 0.4441,  7.2798,  0.0066],
           [25.4555,  0.5107,  6.6482]]]], device='cuda:0', requires_grad=True)
----------------------------------------------------------------------------
fake-INT4_input((XINT - Zero-point)*scale)
 tensor([[[[ 4.2667,  0.0000, 17.0667, 25.6000,  4.2667],
           [12.8000,  4.2667,  4.2667,  0.0000, 17.0667],
           [ 4.2667,  0.0000,  0.0000,  4.2667,  4.2667],
           [ 8.5333,  4.2667,  4.2667,  4.2667,  4.2667],
           [ 4.2667,  4.2667,  8.5333,  0.0000,  4.2667]]]],
       device='cuda:0', grad_fn=<my_grad_scale_funcBackward>)
fake-INT4_weight((WINT - Zero-point)*scale)
 tensor([[[[ 5.8134,  5.8134, 40.6936],
           [ 5.8134,  5.8134,  5.8134],
           [29.0669,  5.8134,  5.8134]]]],
       device='cuda:0', grad_fn=<ThMulBackward>)
```

Fig.s- 3. A toy example of Conv3x3 with weights and activations quantized by INT4 following [7, 8]. Please note that to use GPU Kernel, the INT4 values are recovered to FP32 values with 16 uniform bins.

**Backward Propagation**
**(use mean value as loss for simplicity)**

```python
#check the precision of backward
FP4_loss = torch.mean(FP4_out.view(-1))/10
FP4_loss.backward()
FP32_loss = torch.mean(FP32_out.view(-1))/10
FP32_loss.backward()
np.set_printoptions(precision=5)
```

**Print input gradient (2^[-6,-4,-2,0,2,4,6])**

```python
print("FP32 dL/dX: {}".format(FP32_in.grad.cpu().numpy()))
print("\nFP4-even dL/dX: {}".format(FP4_in.grad.cpu().numpy()))
# to show the fp4 format, set the output to input data prec
```

```
FP32 dL/dX: [[[[0.0064  0.00664 0.43808 0.43169 0.43144]
   [0.01133 0.09246 0.52398 0.51265 0.43152]
   [0.29417 0.38097 0.88636 0.59219 0.50539]
   [0.28777 0.37433 0.44828 0.1605  0.07394]
   [0.28284 0.28851 0.36238 0.07954 0.07387]]]]

FP4-even dL/dX: [[[[0.0625 0.25   1.     1.     1.    ]
   [0.25   0.25   1.     1.     1.    ]
   [1.     1.     1.     1.     1.    ]
   [0.25   1.     1.     0.25   0.25  ]
   [0.25   0.25   1.     0.25   0.0625]]]]
```

**Print weight gradient (2^[-7,-5,-3,-1,1,3,5])**

```python
print("FP32 dL/dW: {}".format(conv32.weight.grad))
print("FP4-odd dL/dW: {}".format(conv4.weight.grad))
# to show the fp4 format, set the output to input data prec
```

```
FP32 dL/dW: tensor([[[[0.5036, 0.6392, 0.8293],
           [0.4790, 0.3068, 0.4559],
           [0.4243, 0.3617, 0.3504]]]], device='cuda:0')
FP4-odd dL/dW: tensor([[[[0.5000, 0.5000, 0.5000],
           [0.5000, 0.1250, 0.5000],
           [0.1250, 0.1250, 0.1250]]]], device='cuda:0')
```

Fig.s- 4. A toy example of Conv3x3 with activation gradients and weight gradients quantized by FP4 as shown in Section 3.1 and 3.3 in the paper. Please note that the output precision of the input activation and weight gradients is set to 4-bit here to demonstrate the aggressively quantized values. In real simulation, the output precision is FP16 as shown in Fig.4 (b).

```
=> creating model 'resnet50o_gs'
=> Model : ResNet(
  (conv1): Conv2dR(3, 64, kernel_size=(7, 7), stride=(2, 2), padding=(3, 3), bias=False |
  mantissa=[9, 9], exponent=[6, 6], backward_mantissa=[9, 9], backward_exponent=[6, 6],
  acc_mantissa=[9, 9], acc_exponent=[6, 6])                        First layer in FP16
  (bn1): BatchNorm2d(64, eps=1e-05, momentum=0.1, affine=True, track_running_stats=True)
  (relu): ReLU(inplace)
  (maxpool): MaxPool2d(kernel_size=3, stride=2, padding=1, dilation=1, ceil_mode=False)
  (layer1): Sequential(
    (0): Bottleneck(
      (conv1): QConv2d_pact_gscale(num_bits_feature=4, num_bits_weight=4, FWD INT4
      act_quant_fn=LearnedClippedLinearQuantization(num_bits=4, clip_val=8.0),
      wei_quant_fn=<function sawb_quantize_param at 0x7efd67837268>, act_clip_init_val=8.0,
      local_gradScale=False, g_data=[23, 8, 0, 3, 2, 5])  BWD dx(FP4) and dw(FP1-5-2)
      (bn1): BatchNorm2d(64, eps=1e-05, momentum=0.1, affine=True, track_running_stats=True)
      (conv2): QConv2d_pact_gscale(num_bits_feature=4, num_bits_weight=4,
      act_quant_fn=LearnedClippedLinearQuantization(num_bits=4, clip_val=8.0),
      wei_quant_fn=<function sawb_quantize_param at 0x7efd67837268>, act_clip_init_val=8.0,
      local_gradScale=False, g_data=[23, 8, 0, 3, 2, 5])
```

Fig.s- 5. An example of a quantized ResNet50 (ImageNet) model (col.4 of Table 2 in the paper), showing the first layer in FP16 and other Conv layers in INT4 forward, FP4 backward GEMM and FP8 update GEMM.

## Appendix B  Quantization Error Analysis

In this section, we provide evidence that, at ultra-low precisions (i.e. FP4), the quantization errors of activation gradients tend to have non-zero expectations, which consequently introduce biases on the output activations as shown in the Eqn. below.

$$
\begin{aligned}
E[dL/dw] &= E[x \cdot (dL/dy + \mathrm{q}_{\mathrm{error}})] \\
&= E[x \cdot dL/dy] + E[x \cdot \mathrm{q}_{\mathrm{error}}]
\end{aligned}
\tag{6}
$$

This bias is difficult to be removed by choosing a fixed threshold for rounding up/down with the nearest rounding scheme, including the one that we choose for minimizing MSE of quantization in the paper. Using the mid-point between two neighbouring exponent levels as the rounding threshold $\theta$, which is $(4^n + 4^{n-1})/2 = 4^n/1.6$, ensures a minimum quantization MSE regardless the distribution of real data. This is shown by scanning the threshold for the gradients in ResNet18 layers in Fig. 2b of the paper, and for the uniform and log distributed random numbers in Fig.s-6 (a). Mathematically, this can be proved by solving the Eqn. (7) and obtaining the Eqn. (8)

$$
\frac{d(\int_{f^{-1}(4^{n-1})}^{\theta}(4^{n-1} - f(x))^2 dx + \int_{\theta}^{f^{-1}(4^n)}(4^n - f(x))^2 dx)}{d\theta} = 0
\tag{7}
$$

$$
f(\theta) = 4^n/1.6
\tag{8}
$$

in which $f()$ denotes the mapping function from an uniform distribution to the data distribution, i.e., for log2-uniform distribution $f(x) = 2^x$.

However, unlike MSE, the mean of the quantization error highly depends on the distribution of data, as shown in Fig.s-6 (b), where the uniformly distributed data has zero mean as expected, but the log-uniform distributed data does not. We also verify the non-zero mean of the quantization error in the real gradient data in Fig.s-7. As seen for all layers in the last block of ResNet18 (ImageNet), the quantization error's mean is non-zero at the chosen threshold of $4^N/1.6$. Also, there's no fixed threshold that can achieve zero mean for all four gradients in the figure due to their different distributions. This observation motivates us to minimize the MSE by choosing the mid-point of $4^N/1.6$ in Fig.2 (c), and use other methods, such as the Two-Phase Rounding to minimize the non-zero mean. It should be noted that the quantization bias is much less significant for a higher precision because data could be considered uniform in the denser bins of higher precisions—just like the uniform numbers in Fig.s-6 (b)—which results in approximately zero-mean when choosing the mid-point as the threshold.

Fig.s- 6. the MSE (a) and Mean (b) of 4-bit quantization on log-uniform and uniform distributed random data with scanned rounding thresholds

Fig.s- 7. the Mean of 4-bit quantization error on real gradients of the last block of ResNet18 (ImageNet) with scanned rounding thresholds

## Appendix C   Hardware performance evaluation

In this paper, we focus on comparing the hardware performance on the three GEMMs of DNN model training (forward, backward and update) and providing a clean comparison among different combinations of precisions. We use compute density (area of digital logic) as the metric for comparing arithmetic engines. In accelerators designed in modern CMOS technologies with fully-pipelined micro-architectures, compute density is a good proxy for power efficiency of the designs[9]. Thus, to the first order, the compute density improvements can also be considered power efficiency improvements (usually measured in tera-operations/sec/W).

For the 4-bit backpropagation, a multiply-accumulate (MAC) unit that performs 4-way $INT4 \times FP4$ products consumes 55% of the area of a baseline FP16 FPU while providing $4\times$ the compute throughput, yielding a total compute density improvement of **~7×**. Our area projections are based on insights derived from our previous low-precision FPU design [10]. Compared to the baseline FP16 FPU, we expect the $INT4 \times FP4$ compute unit to have simple shift-based multipliers since the FP4 numbers are powers of 2, no addend aligners, narrower adders and a less complex normalizer. Additionally, the simpler datapath requires fewer execution stages for the same operating frequency targets, yielding latch area savings. Employing the same methodology, we also obtain the performance for other combinations of precisions normalized to FP16 FPU's, as listed in the table s-1.

Table s- 1. The performance improvement of low-precision MAC units

| Prec1 | Prec2 | Performance |
|-------|-------|-------------|
| FP16 | FP16 | 1.0 (reference) |
| INT4 | INT4 | $\sim 8\times$ |
| INT4 | FP4 | $\sim 7\times$ |
| INT4 | FP8 | $\sim 2\times$ |
| FP8 | FP8 | $\sim 2\times$ |

With the table s-1, we obtain the total performance of executing all three GEMMs of training in different precisions against the FP16 training, as shown by Fig.6c in the paper. For example, the full 4-bit training is $7.3\times$ higher in performance than the FP16 training, as estimated by $\frac{3}{1/8(=INT4 \times INT4) + 2/7(=INT4 \times FP4)}$; and the hybrid 4-bit training is $3.9\times$ higher in performance, as estimated by $\frac{3}{1/8(=INT4 \times INT4) + 1/7(=INT4 \times FP4) + 1/2(=INT4 \times FP8)}$.

For evaluating the performance of selectively using FP8 Conv 1x1 layer in ResNet50 (ImageNet), we assume the FLOPs of the Conv 1x1 layer is 4/9 times of the Conv 3x3 layer as the former has 1/9 of kernel sizes and 4 times of feature maps of the latter.

# Appendix D    Model training Details

## D.1    CIFAR10

The architectures of six models used on CIFAR10 dataset, i.e. VGG16 [11], GoogLeNet [12], ResNet18, ResNet50, ResNet101 [13] and DenseNet121 [14], are all adapted from their original published papers. The same standard pre-processing, i.e. RandomCrop(), RandomHorizontalFlip() and Normalize(), are used for training all the models. All the models are trained for 160 epochs using standard momentum SGD with batch size of 128, initial learning rate of 0.1 and momentum of 0.9. A step learning rate schedule is used with 0.1 decay at epoch 82 and 122. For VGG16 and ResNet18, no weight decay is applied, while the rest networks share the same weight decay of 1e-4. All converging curves are shown in Fig.s-8,9,10,11,12,13 and 14.

Fig.s- 8. VGG (CIFAR10) model convergence with FP32 (baseline) and with gradients in Two-Phase-Rounding full FP4 and partial FP4 (backward GEMM in FP4 and update GEMM in FP8), while using INT4 for weights and activations (WA-INT4).

Fig.s- 9. GoogLeNet (CIFAR10) model convergence with FP32 (baseline) and with gradients in Two-Phase-Rounding full FP4 and partial FP4 (backward GEMM in FP4 and update GEMM in FP8), while using INT4 for weights and activations (WA-INT4).

Fig.s- 10. ResNet18 (CIFAR10) model convergence with FP32 (baseline) and with gradients in Two-Phase-Rounding full FP4 and partial FP4 (backward GEMM in FP4 and update GEMM in FP8), while using INT4 for weights and activations (WA-INT4).

Fig.s- 11. ResNet50 (CIFAR10) model convergence with FP32 (baseline) and with gradients in Two-Phase-Rounding full FP4, partial FP4 (backward GEMM in FP4 and update GEMM in FP8) and FP4 with 1x1 Conv layers in FP8, while using INT4 for weights and activations (WA-INT4).

Fig.s- 12. ResNet101 (CIFAR10) model convergence with FP32 (baseline) and with gradients in Two-Phase-Rounding full FP4, partial FP4 (backward GEMM in FP4 and update GEMM in FP8) and FP4 with 1x1 Conv layers in FP8, while using INT4 for weights and activations (WA-INT4).

Fig.s- 13. DenseNet121 (CIFAR10) model convergence with FP32 (baseline) and with gradients in Two-Phase-Rounding full FP4 and partial FP4 (backward GEMM in FP4 and update GEMM in FP8), while using INT4 for weights and activations (WA-INT4).

Fig.s- 14. MobileNetV2 (CIFAR10) model convergence with FP32 (baseline) and with gradients in Two-Phase-Rounding full FP4 and partial FP4 (backward GEMM in FP4 and update GEMM in FP8), while using INT4 for weights and activations (WA-INT4).

## D.2 ImageNet

### D.2.1 ResNets

We adapt ResNets v1 (including ResNet18 and ResNet50) from the models of torchvision[15] and the standard pre-processing of ImageNet ILSVRC2012 dataset. We use a learning rate of 0.1 with a 0.1 decay of epoch 30, 60 and 90 and trained to 100 epochs. For the optimizer, we use non-Nesterov SGD with a momentum of 0.9 and a weight decay of 1e-4. The BatchNorm running mean and variance momentum are set to 0.1. The minibatch size is 256 over 8 V100 GPUs. Eventually, we obtain 69.40% and 76.48% for the baseline accuracies of ResNet18 and ResNet50, as shown in Fig.s-15 and Fig.s-16.

Fig.s- 15. ResNet18 (ImageNet) model convergence with FP32 (baseline) and with gradients in Two-Phase-Rounding full FP4 and partial FP4 (backward GEMM in FP4 and update GEMM in FP8), while using INT4 for weights and activations (WA-INT4).

Fig.s- 16. ResNet50 (ImageNet) model convergence with FP32 (baseline) and with gradients in Two-Phase-Rounding full FP4 and partial FP4 (backward GEMM in FP4 and update GEMM in FP8), while using INT4 for weights and activations (WA-INT4).

### D.2.2 AlexNet

We adapt the AlexNet model from torchvision[15] and the standard pre-processing of ImageNet ILSVRC2012 dataset. We use the learning rate of 0.01 with a 0.2 decay at epoch 20 and 30 and trained up to 50 epochs. For the optimizer, we use non-Nesterov SGD with a momentum of 0.9 and a weight decay of 1e-4. The minibatch size is 256 over 8 V100GPUs. Finally, we obtain the accuracy of $57.56\%$ matching the standard[16], as shown in the Fig.s-17.

Fig.s- 17. Alexnet (ImageNet) model convergence with FP32 (baseline) and with gradients in Two-Phase-Rounding full FP4 and partial FP4 (backward GEMM in FP4 and update GEMM in FP8), while using INT4 for weights and activations (WA-INT4).

### D.2.3 MobileNetV2

We adapt a PyTorch implementation of MobileNetV2 and the standard pre-processing [17] of ImageNet ILSVRC2012 dataset. We use learning rate 0.05 with a cosine learning schedule finished in 150 epochs. For optimizer we use SGD with Momentum of 0.9 and weight decay of 4e-5. The BatchNorm running mean and variance momentum is set to 0.1. For our purpose of studying quantization impact, the weight averaging of 0.999 adopted by the original paper[18] is not used. The minibatch size is 256 over 8 V100 GPUs. Finally, we obtain the accuracy of 71.85% matching the standard[18], as shown in the Fig.s-18.

Fig.s- 18. MobileNetV2 (ImageNet) model convergence with FP32 (baseline) and with gradients in FP32, Two-Phase-Rounding full FP4, partial FP4 (backward GEMM in FP4 and update GEMM in FP8), and FP32 while using INT4 for weights and activations (WA-INT4).

### D.3 LSTM PTB

We adapt the two-layer LSTM model with the medium size (with a hidden dim of 650) from Pytorch Examples[19] on PennTreeBank dataset with vocabulary size of 10,000. We adopt a static learning rate 1.0 with a 0.8 decay factor after the Epoch 6. The minibatch size is 20 and the sequence length is 35. Eventually we obtain a Validation and Test perplexity of 86.41 and 83.34, respectively, as shown in Fig.s-19. It should be noted that for the 4-bit forward, we shift the bins of WA-INT4 by one half of the bin to the left for aligning the zero in INT4 with the absolute zero value (also reducing the total level from 16 to 15 to remain symmetric around the zero level)—this is important to LSTM networks and easy to implement in the hardware. Following the convention of quantization[6], we keep the first embedding layer and the last decoding (FC) layer of the speech model in FP16 (1-6-9).

Fig.s- 19. the 2-layer LSTM (PTB) model convergence in the Validation Perplexity with FP32 (baseline) and with gradients in Two-Phase-Rounding full FP4 and partial FP4 (backward GEMM in FP4 and update GEMM in FP8), while using INT4 for weights and activations. *The WA-INT4 format is shifted by one half of the bin to the left for aligning the zero in INT4 with the absolute zero value.

## D.4 Transformer

We adapt the implementation of FairSeq [20] and the setup of the Transformer Base model in the repository on the WMT 14 En-De Translation task. We used the Adam optimizer. Detailed parameters can be found in the repository of FairSeq [21]. To calculate BLEU score vs. Epoch we used the script in the repository on the checkpoint generated at each epoch with beam 4, length penalty of 0.6, and removebpe option after compound splitting [22]. Eventually we obtained BLEU score  27.5 at Epoch 32 with 120,000 updates as shown in Fig.s-20, in line with the publication [23].

Fig.s- 20. the Transformer-base model convergence on WMT14 En-De translation task with FP32 (baseline) and with gradients in Two-Phase-Rounding full FP4 and partial FP4 (backward GEMM in FP4 and update GEMM in FP8), while using INT4 for weights and activations.

### D.5    Speech Model

We adapt the acoustic LSTM model of IBM speech [24] that contains 4 bi-directional LSTM layers and 2 fully-connected layers in the main network. Each LSTM layer contains 1024 cells with 512 on each direction. On top of the LSTM layers, there is a linear bottleneck layer with 256 hidden units followed by a softmax output layer with 32K units corresponding to CD HMM states. The LSTMs are unrolled 21 frames and trained with non-overlapping feature subsequences of that length. The 300-hours switchboard data (SWB300) set is used to train network. The training set consists of 262 hours of Switchboard 1 audio with transcripts. The test set is the 2.1-hour switchboard (SWB) data from 40 speakers. The batch size is 128. Bigram language models (LMs) are used in decoding and acoustic weight is chosen as 0.05. The full precision Word Error Rate (WER) we obtained closely matches the one reported in [24]. Same as the LSTM (PTB) task, for the 4-bit forward, we shift the bins of WA-INT4 by one half of the bin to the left. Following the convention of quantization[6], we keep the first embedding layer and the last FC layer of the speech model in FP16 (1-6-9).