[Reviews · NeurIPS 2020]

Review 1

Summary and Contributions: The authors propose novel techniques to enable 4-bit (floating-point) quantized training for DNN training. The first technique is radix-4 fp4, which is a 1-3-0 (sign-exponent-mantissa) representation. This was empirically tested to perform better than other fp4 schemes. The second technique is GradScale, which is adaptive per-layer loss scaling (similar to NVIDIA's fp16 mixed-precision efforts which use a global adaptive loss scale except per-layer). The third technique is two phase rounding (TPR), where the incoming gradient for a layer is quantized in two different ways, one to calculate dL/dw (weight update) and another to calculate dL/dx (back-propagated gradient). The two ways of rounding are even (round to 2^even) and odd (round to 2^odd). Quantization introduces error with a bias, and TPR can help cancel out the this bias by rounding in two opposite directions. The authors show how quantization bias is a key problem in the batch norm layers because the variation in mean between different quantized batches can be large. A combination of TPR and running the 1x1 convs in fp8 can address this issue for ResNets on ImageNet. The hardware quantization scheme proposed uses int4 input activations, int4 weights, and fp4 input gradients. MAC outputs are accumulated in fp16. Experimental data on (1) a variety of CIFAR-10 networks; (2) AlexNet, ResNet-18, and ResNet50 on Imagenet; (3) a 2-layer LSTM and a 4-layer biLSTM show that with these techniques fp4 training achieves within ~1% of the fp32 accuracies.

Strengths: The strongest part of the paper is the study on how TPR affects the BN stats in the batchnorm layers. This is both very insightful for the reader as well as great evidence that TPR is doing what the authors theorized it would do. Excellent and compelling research.

Weaknesses: I'm not sure how TPR physically works in hardware. Say I have 2 exponent bits and can represent 4 values (00, 01, 10, 11). Does Even rounding mean that (00->2^0, 01->2^2, 10->2^4, 11->2^6), and similar for Odd rounding? That is, we map bit representations directly to the allowed exponents? This means that you would need dedicated MAC units to interpret Even/Odd rounded numbers. The MAC units that runs Even fp4 can't be used for the Odd or regular fp4. This is going to have huge effects on how you map the network to hardware and possibly throughput. In Section 4.2, it is stated that Var(dL/dx) becomes larger as gradients propagate through the network, which motivates the use of fp8 in the Conv1x1s. But Figure 6a shows that the grads get smaller from layer 50 to layer 1. Furthermore, the grads are smaller when using pure fp4 than fp4+fp8. I'm not sure if I understood this part. Figure 6a seems to tell me that the heavily quantized networks have a problem with vanishing gradients. Claim 5 in the list of technical contributions say that fp4 is used for the "majority" of computations and fp8 for the Conv1x1 layers. This is true for a baseline ResNet, but many modern networks have adopted separable convolutions, where the 3x3 convs are depthwise. In these networks the 1x1s are the majority of computations. EDIT: In the rebuttal, the authors clarified how even/odd rounding works in hardware and how on some networks, the 1x1s can be done in fp4. It bothers me that which part of the network can be computed in fp4 seems to be trial-and-error, but this is still an improvement.

Correctness: One major issue for me is that evaluation is done mostly on small CIFAR-10 models, which a few ImageNet ResNets and small LSTMs. The data does not indicate these techniques can enable fp4 training for networks that are close to SOTA-size. Indeed, fp8 had to be used to support ResNet-50 on ImageNet and the biLSTM. The claims about 7x hardware area and power savings were not backed up in any way (no experiments or citations). Still, I believe the numbers are in the right ballpark, and it's very difficult to evaluate this since the proposed scheme is very non-standard (int4 weights and activations, mixed fp4/8 gradients). EDIT: In their rebuttal, the authors demonstrated FP4 training with MobileNetV2 on ImageNet and Transfomer-Base. There is still insufficient detail on the hardware needed to support two-phase rounding. I'm willing to bump the score to a 7.

Clarity: Yes

Relation to Prior Work: Yes, the novelty is clear. Stochastic quantization in binary networks should be mentioned as a prior work for TPR (see https://arxiv.org/abs/1602.02830).

Reproducibility: Yes

Additional Feedback:


Review 2

Summary and Contributions: The paper demonstrates the DNN training with 4-bit weights, activations, and gradients. The author combined a collection of approaches from previous work such as PACT, SWAP, and FPSC and proposed approaches such as radix 4 FP4, two-level rounding quantization, hybrid numerical formats (FP4, FP8, and INT4). The efficiency of the proposed method is evaluated on CIFAR-10, Imagenet, and PTB datasets. It shows a negligible accuracy loss compared to FP32. Moreover, the hardware complexity of DNN training with 4-bit parameters are estimated based on area density.

Strengths: Demonstrating the DNN training for the very first time and achieving accuracy close to FP32.

Weaknesses: The goal of low-precision training is to reduce hardware complexity while also tolerating accuracy degradation. The paper achieved the second goal, but it is not obvious the DNN with 4-bit training is less complicated than DNN with 8-bit training [1,2]. The overhead of collection approaches such as PACT, high radix representation, two-time rounding, layer-wise scaling, and et al. is enormous. The paper should compare the overhead of DNN with 4-bit training in detail with 8-bit training [1,2]. If the overhead becomes more than 8-bit training, there is no advantage to using 4-bit training as long as the approach's cost is reduced.

Correctness: The author claims that the hardware complexity of converting a number between radix 4 and radix 2 is negligible. However, this claim is not correct. The conversion needs comparison with number 3, and it is complex and the number of times that conversion is an essential factor. The author should elaborate on the hardware complexity of radix 4 and the number of conversions in the paper. [1] Sun, Xiao, et al. "Hybrid 8-bit floating point (HFP8) training and inference for deep neural networks." Advances in Neural Information Processing Systems. 2019. [2] Cambier, Léopold, et al. "Shifted and Squeezed 8-bit Floating Point format for Low-Precision Training of Deep Neural Networks." arXiv preprint arXiv:2001.05674 (2020).

Clarity: The paper is not organized well. It's tough to read and follow the different algorithms presented in the paper. Many information in the paper and appendix, such as explaining the software is unnecessary. The organization of the paper should be improved.

Relation to Prior Work: Yes

Reproducibility: Yes

Additional Feedback: The author addressed my feedback in the rebuttal. As a suggestion, It would be good to add a figure or table to compare the hardware complexity and previous work [1,2]. Moreover, I also agree with other reviewers. The title could be changed as a Hybrid 4-bit rather than 4-bit training. [1] Sun, Xiao, et al. "Hybrid 8-bit floating point (HFP8) training and inference for deep neural networks." Advances in Neural Information Processing Systems. 2019. [2] Cambier, Léopold, et al. "Shifted and Squeezed 8-bit Floating Point format for Low-Precision Training of Deep Neural Networks." arXiv preprint arXiv:2001.05674 (2020).


Review 3

Summary and Contributions: This paper introduces a few techniques, building on previous works (.e.g, PACT), that nearly achieve full precision accuracies. It motivates its methods by highlighting the growing size and complexity of DNNs, which necessitate more compression (e.g. low bitwidth) techniques during training process. The claimed technical contributions include a radix-4 FP4 format used to represent gradients, a gradient scaling technique, GradScale, to be used with this format, a two-phase rounding (TFR) method to reduce the rounding error on the gradients with an accompanying theoretical study, and an empirical analysis for still running some layers in higher precision (FP8 for pointwise convolutions). Additionally, they include a small analysis on the effect of this FP4 format on the gradient error, and in the supplementary material, a discussion on the potential performance increase (in terms of compute density) expected from their methods. They evaluate on a range of CNNs and LSTMs on CIFAR10, ImageNet, and PTB and come within ~1% of full-precision accuracy.

Strengths: Their motivation and framing for their low-precision training was strong: there is a quadratic dependency of precision on system throughput, 16-bit training support has already been introduced into commercial accelerators, and 8-bit training techniques have been shown effective in many cases. It makes sense that the next goal should be a system that fully uses 4-bit compute. The ablation studies were detailed and motivated a lot of the technical contributions. Many models were tested, across different tasks, including fairly large models with DenseNet121, which is particularly important in this work since it should be marketed toward very large models.

Weaknesses: Ablation study Figure 6a: 1x1 convs often represent a large portion of the FLOPs in networks, and for that reason works like ShuffleNet break them into group convolutions. Is there anything fundamental about the 1x1 conv to justify putting it in FP8? What is the effect of putting just the 3x3 layers in FP8? Especially if the 1x1 convs include conv in the identity branch, most of the conv layers would now be in FP8. Perhaps it should be stated what percentage of gradients are in 8-bit, although this may be captured in the estimated performance data. Edit: After the author feedback, I still believe that it is misleading to label the method as fully 4-bit when a significant number of layers are cast in FP8. I expect there should be some relevant related works related to reducing error in rounding. The interesting part of this problem is that the value is used twice and you have chosen two (slightly) different formats for each. This seems to increase the precision at a given bitwidth, i.e. two FP4 numbers can represent the same number of values as a FP5 number, but there should be a tradeoff in handling the extra formats. Edit: The authors reasonably addressed this concern in their feedback. Motivation for radix-4 system: I understand why radix-4 spans a larger range and empirically it is shown to perform better than radix-2, but it isn't clear to me why radix-4 is optimal. Would higher radix values also work? The FP4 format seems like just a 4-bit logarithmic system (which was briefly mentioned in the paper) and in the related work 5-bit logarithmic systems are referenced. It seems like the novelty here is applying to gradients. Is there a fundamental contribution of the format outside of decreasing the radix? It also seems generous to include the FP8 1x1 convs as a technical contribution, especially since there wasn't much theoretical contribution made as to why the 1x1 convs are particularly sensitive.

Correctness: The claims and methods in the paper seem correct as presented.

Clarity: The figures were very important to the presentation of the paper, but at times they were complicated, miss pieces, and somewhat confusing. Figure 1 should have more useful titles instead of the hyperparams, which are very similar across figures. These hyperparams might be more appropriate in the caption or discussion. Also, Figure 1b, for example, is missing a title. Many of the parts in Figure 1 aren't referenced until much later in the paper, which requires the reader to frequently scroll while reading. Figure 3b is key in motivating GradScale, but it could be made clearer, possibly by using only some of the layers. A very minor point is that different font sizes are used in adjacent figures.

Relation to Prior Work: This work introduces previous papers that address low-precision training and distinguishes itself by claiming to be the first to reduce the whole training process to 4-bits (including gradients). It clearly states that it builds on top of PACT, SAWB, and FPSC.

Reproducibility: Yes

Additional Feedback: GradScale gives different scale factors for individual layers citing the different distributions among them. It is known that different channels have different distributions of activations and weights, and presumably this leads to different distributions of gradients. Are there any additional experiments extending the gradient quantization to the channel-level like other quantization techniques? Figure 4 is well made and introduces TPR clearly, yet from the figure along, the benefits of TPR to accuracy isn't clear. For example, if you are at integers, TFR should introduce more error compared to rounding to integers directly. If you are at half-integers, it should erase the error completely. Since the error should be continuous, there must be points close to the integers that also increase the error to some extent, yet the regions with error cancelation seem to cover these regions too. By symmetry, it seems like it would lead to a 50% chance of cancelling error and 50% chance of increasing it. Edit: The author feedback cleared this up. I originally incorrectly compared the 4-bit even/odd rounding to a 5-bit rounding.


Review 4

Summary and Contributions: The authors present an analysis of existing approaches to low-bit training of neural networks and present improvements and new techniques when moving to even lower, 4bit training. Theoretical analysis and experimental validation paint a convincing picture. ======= I have read the rebuttal and discussed with the other reviewers and AC

Strengths: This paper tells a convincing story about challenges with existing approaches to 4bit training, analysis of problems and presents strong evidence for its proposed solutions. The significance of the work is high if the required hardware components will eventually be realised.

Weaknesses: As the authors remark themselves, the general applicability of the proposed approach to 4bit training might not be given and instead need to be validated for the individual NN architecture / dataset combination. As such, this paper is a promising step into the right direction.

Correctness: Yes

Clarity: The paper is written clearly. Some minor details include: line 30: "has been shown" Line 34: "backward and update phase entirely in 4-bit representation" Line 240: "residual block" Line 246: "Experimental evaluation" or something like that. The title seems incomplete The authors frequently use abbreviations such as "we've", which I believe is only used in colloquial writing (or speaking). In formal language, "we have" is written out. (Not a native speaker though)

Relation to Prior Work: Yes

Reproducibility: Yes

Additional Feedback: In figure 1(b), it is not possible to see which gradient distribution from ep0 and ep100 belongs to the same layer. It would be interesting to understand how an individual layer changed over the course of training. Finally, the broader impact study shows no discussion of potential negative ethical and societal implications of this work. One such implication I could imagine is that there is of yet no study (as far as I know) about how low-bit training introduces bias into the final model. Admittedly far fetched, maybe the authors have more suggestions.

[Author Response · NeurIPS 2020]

**We thank all four reviewers for their valuable feedback. All suggestions will be included in the final draft.**

**Reviewer1** On TPR hardware: The reviewer's understanding of even/odd phase is correct—but dedicated MAC units are not needed to handle the additional odd phase. FP4 numbers in the odd phase are simply $0.5\times$ of those in the even phase. Thus, the odd-phase final MAC result is also $0.5\times$ and can therefore be efficiently computed by subtracting 1 from the exponent of the final FP16 result using the same MAC engine. On Sec.4.2: Thank you for finding the typo. To clarify, it is the gap between TPR's Var(dL/dx) and FP32's Var(dL/dx) that increases from layer50 to layer1. Our results on FP4 and "FP4+FP8" indicate that quantization causes the loss of gradient variance in Fig.6a. Also in [35], vanishing gradients are reflected by the loss of gradient variance—so we'll look to study this correlation further. On Depthwise Conv (DW): The variance analysis and the hybrid approaches are not limited to Conv1x1 and can be applied to compact networks as well. In fact, while analyzing MobileNetV2 with DW, we found it optimal to use 4b for Conv1x1 and follow the FP8 work recommendations to use FP16 for DW layers[15]. FP16 DW does not impact latency much since it is <3% of total FLOPs and is hard to accelerate due to limited data reuse. For MobilenetV2, we achieve 1.2% degradation on CIFAR10 and 3.0% degradation on ImageNet using FP4. These new accuracy results are especially promising—given that just INT4 inference in MobilenetV2 remains challenging [23,24]. On SOTA models: To our knowledge, this is the first time a SOTA-size model has ever been trained in 4b without major convergence issues. To show the robustness of our approach over a larger model set, we trained a large NLP task with transformer_base model on WMT De-En and obtained a good BLEU score with 2 points off the baseline (25.4 vs 27.5). Overall, we believe this work lays a solid foundation upon which future FP4 innovations can be added on to address the remaining accuracy losses. On hardware area/power: The area projection is based on insights derived from multiple previous AI Hardware and Low-precision FPU designs (published in Symp. on VLSI circuit). In our estimates, a MAC unit that performs 4-way INT4$\times$FP4 inner products consumes 55% of the area of the FP16 FPU while providing $4\times$ throughput, yielding a total compute density improvement of $7.3\times$. Compared to FP16 FPUs, the 4b unit has simpler shift-based multipliers thanks to the power-of-2 FP4 numbers. It also benefits from the absence of addend aligners, narrower adders, and a simpler normalizer. Additionally, the simpler datapath requires fewer executing stages and saves flip-flop area.

**Reviewer2** On 4b overhead: We've estimated that the overheads for 4b training to be <5% of the total GEMM ops. In back-propagation, all collection approaches cost $\mathcal{O}(1)$ FLOPs/gradient element. Compared with HFP8[15] and S2FP8, our FP4 conversion is actually simpler due to the absence of mantissa rounding—and can be done with two back-to-back instructions(1 MUL+1 custom bit OP). The overhead of statistics collection for layer-wise scaling is also comparable to that in S2FP8. Specifically, we expect ~10 FP16 FLOPs/gradient for PACT BWD(2), Radix Conversion(3), Two-phase Rounding(3), and Layer-wise Scaling(2) overheads. These overheads are much smaller than $\mathcal{O}(k_i \times k_j \times channel)$/gradient in convolution GEMMs (e.g. In ResNet50, the effective GEMM FLOPs is 642 per gradient element). Therefore, **with the majority of FLOPs spent on GEMM, 4b training retains significant advantage over HFP8 and S2FP8 training due to the throughput and power & area boost in going from 8b to 4b GEMM**. With additional optimization from our compiler [published in IEEE Micro], 4b ResNet50 training can yield at least 60-80% higher throughput vs. HFP8 training along with a 42% area and power saving. On conversion hardware: The conversion between radix-2 and radix-4 is remarkably simple for FP4. Due to the absence of mantissa bits, this can be done through simple bit operations on exponenst - which rounds down the FP16 exponent to even or odd value (and handles overflow and underflow) through common clipping and rounding operations.

**Reviewer3** On Conv1x1: Our mixed precision approach is not limited to only Conv1x1, which is chosen in the context of ResNet50. For compact models like ShuffleNet and MobileNet, Depthwise convolution layers will be in higher precision (see line 8). Although the Conv3x3 layers have $2\times$ FLOPs than the bottom-conv1x1, they yield less accuracy improvements when cast in FP8 (74.7 vs. 75.0). While the performance is determined by FLOPs instead of gradient size, bottom-conv1x1 has $4\times$ output gradients than conv3x3 and top-conv1x1, explaining its efficiency in accuracy gain when cast in FP8. Tradeoff of TPR: Please see line 2. On optimal radix: Radix-2 does not have enough range for gradients. Our experiments with radix-8 on BWD and/or UPDATE GEMM show that training cannot converge due to poor representation. In addition, for hardware implementation, radix-8 cannot adopt TPR to enhance its representation due to the non-integral exponent of dividing $2^3$ into two phases. On contribution of format: In the context of radix-4, we propose efficient rounding schemes that minimize the MSE of gradient quantization. In addition, we propose the hybrid use of INT4 for FWD and FP4 for BWD. On channel-level quantization: We limit the quantization granularity to per layer for hardware efficiency. If each output-channel has a different scale, reduction over the channel dimension will be hardware inefficient. On cancellation: Fig.4a shows that the same gradient will most likely be quantized towards opposite directions in each phase. This is important to get a lower expected quantization error $\mathbb{E}(Q_{err})$, which is the sum of mean errors over the entire ensemble from each phase **divided by 2**. The cancellation contributes to a lower average expected value but individual values could certainly be skewed higher or lower.

**Reviewer4** On empirical validation: We agree with the reviewer and would like to point that we have also evaluated other models (please refer to lines 12&16). In future, we plan to utilize AutoML techniques to make FP4 broadly applicable. On bias: The numerical bias caused by quantization is random and not expected to result in fairness issues — especially since it acts at lower levels than the data/target/model choices and is therefore not directly susceptible to human intervention. We do plan to validate this further as part of future work.

[Meta-Review · NeurIPS 2020]

Fast training and model compression are important issues when applying machine learning techniques in practice. The proposed 4-bit training method in this paper is novel. The empirical experiments are comprehensive and the results are promising. A minor issue is that it does not seem very clear how hardware could well support this method. Please add some discussions on this in the final version.